# No Imputation Needed: A Switch Approach to Irregularly Sampled Time Series

## Abstract

Modeling irregularly-sampled time series (ISTS) is challenging because of missing values. Most existing methods focus on handling ISTS by converting irregularly sampled data into regularly sampled data via imputation. These models assume an underlying missing mechanism, which may lead to unwanted bias and sub-optimal performance. We present `SLAN` (Switch LSTM Aggregate Network), which utilizes a group of LSTMs to model ISTS without imputation, eliminating the assumption of any underlying process. It dynamically adapts its architecture on the fly based on the measured sensors using switches. SLAN exploits the irregularity information to explicitly capture each sensor's local summary and maintains a global summary state throughout the observational period. We demonstrate the efficacy of SLAN on two public datasets, namely, MIMIC-III and Physionet 2012, for the in-hospital mortality prediction task. The code will be publicly made available upon acceptance of the manuscript.

## 1 Introduction

An irregularly sampled time series (ISTS) is a multivariate time series recorded at inconsistent or non-uniform time intervals. Such data can be found in various fields dealing with complex generative processes, like meteorology (Mudelsee, 2002), seismology (Ravuri et al., 2021), user social-media activity logs (Zeng & Gao, 2022), e-commerce transactions (Wu et al., 2013), epidemiological and clinical research (Shrive et al., 2006; Yadav et al., 2018). The cause of missingness in ISTS relates to unobserved data (Ma & Zhang, 2021). Thus, modeling applications concerning ISTS is challenging. This work focuses on modeling ISTS in the clinical domain since it is a well-established and critical application.

Many methods handle ISTS by filling missingness via imputation converting ISTS (see Fig. 2(b)) to regularly sampled time series (Fig. 2(a)), assuming an underlying missing mechanism (Ipsen et al., 2020). Imputation is the process of filling up missing values with estimated values whenever input is not observed. The imputation can be performed via forward filling, mean (Che et al., 2016), interpolation (Shukla & Marlin, 2018), model-based technique (Lim et al., 2021b), etc. However, any form of imputation may alter the data's original nature with artificial approximation, leading to unwanted distribution shifts (Zhang et al., 2023). This may introduce a bias, resulting in sub-optimal performance. We show this empirically in our findings, where the performance of our model (SLAN) consistently outperforms the imputation-based models (see Table 2). We further validate this by introducing additional missing observations (Figure 1) and comparing SLAN with the best imputation model (IPNets), where SLAN outperforms IPNets in all scenarios.

We argue that learning the imputation task is challenging because of the underlying missing mechanism and may not be required for the downstream task. Some non-imputation methods (Horn et al., 2020; Vaswani et al., 2017) exist in the literature but do not properly exploit the temporal structure of missing values. It is worth noting that ISTS datasets are not simply incomplete but contain informative missingness (Rubin, 1976). Therefore, specialized methods are required to handle such missingness in a meaningful way.

We present *Switch LSTM Aggregate Network* (SLAN), which utilizes a group of LSTMs to handle ISTS. Our proposed model exploits the irregularity information of the ISTS to maintain the local summary state for each observed time series. In most practical situations, these time series are coming from sensor measurement.

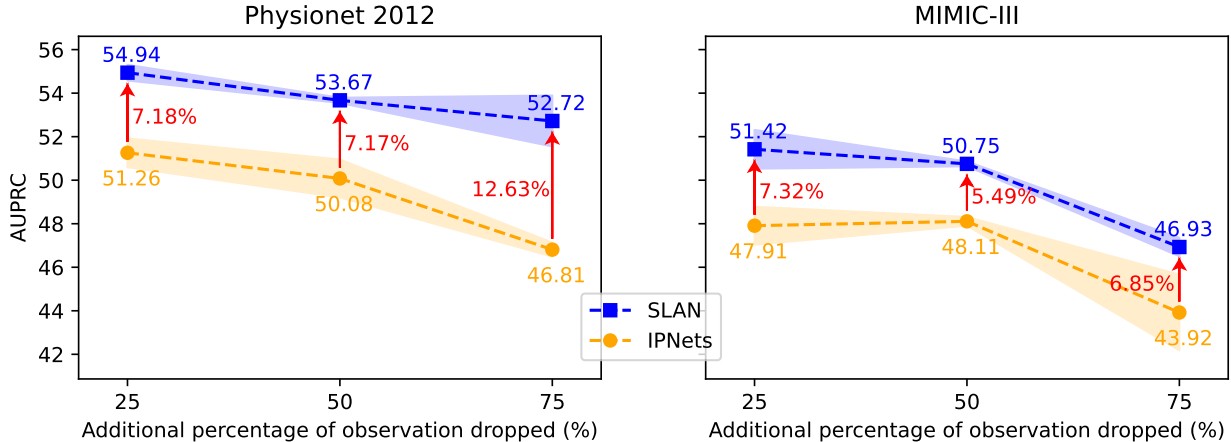

Figure 1: AUPRC of SLAN vs IPNets on P-12 and M-3 datasets with a drop of 25%, 50%, and 75% observed data. The red arrows show the % increase in the AUPRC of SLAN compared to IPNets with % increased value mentioned in the red-colored number. The blue and orange colored numbers represent the AUPRC of SLAN and IPNets, respectively.

The model is equipped with a switch layer that enables it to adapt to the input order of measured sensors dynamically. SLAN maintains a global summary state, aiding each LSTM with summarised information throughout the observational period. We show the efficacy of SLAN on two widely used public clinical datasets: MIMIC-III (Johnson et al., 2016) and PhysioNet 2012 (Goldberger et al., 2000) for the in-hospital mortality prediction, i.e., binary classification task. The contributions of our work are: (1) We propose a simple and effective switch layer that dynamically changes the SLAN architecture to handle ISTS, thus eliminating the need for imputation. (2) We introduce a global summary state enriched with the information from each sensor. (3) We maintain a local summary state for each sensor, aided by other sensor information.

## 2 Related Works

### 2.1 Imputation-Based Models

Previous works such as (Marlin et al., 2012; Harutyunyan et al., 2017; Rasmussen & Williams, 2003) and IPNets (Shukla & Marlin, 2018) propose to convert ISTS to regularly sampled time series by temporal discretization. They perform imputation using a probabilistic model (Marlin et al., 2012) or interpolation (IPNets). The main drawback of these methods is that they require either ad-hoc choices such as width and the aggregation function (Marlin et al., 2012) or a predefined nonlinear form assuming that missingness is at random and thus may induce bias. GRU-D (Che et al., 2016) impute the missing input by employing learnable decay on the global mean and the last measured value. ViTST (Li et al., 2024) transforms ISTS into line graphs by interpolating the missing values. The graphs are then fed to a vision transformer. Overall, these models perform imputation, assuming an underlying missing mechanism, which may lead to unwanted bias. We present a detailed discussion of the imputation mechanism of GRU-D, IPNets, and ViTST in section A of the Appendix. Our study includes GRU-D, IPNets, and ViTST as a baseline model because of their demonstrated efficacy in modeling ISTS.

### 2.2 Non-Imputation Models

Recent approaches have also explored learning directly from the ISTS data without any form of imputation. Transformer (Vaswani et al., 2017) based models have been widely used for modelling time series data, including ISTS. These approaches mainly replace the positional encoding with an encoding of time and model sequences using self-attention and concatenate it with the input representation. The main drawback

of these methods is the permutation-invariant nature of self-attention, which may be problematic in capturing dependencies within each time series. SeFT (Horn et al., 2020) proposed to learn from ISTS via set-based data representation, treating time series as an unordered set of measurements. However, the order-invariant nature of set representation fails to capture the irregular information, which is order-variant and increases with time. Raindrop (Zhang et al., 2021) proposes a graph neural network to learn a sensor dependency graph. Raindrop leverages inter-sensor dependency to train latent embedding. However, Raindrop may not exploit the irregularity information of the sensors. Recent methods like CoFormer (Wei et al., 2023) utilize a transformer-based encoder for temporal-interaction feature learning and IVP-VAE (Xiao et al., 2024) models ISTS with continuous processes whose state evolution is approximated by initial value problems.

## 3 Problem Formulation

### 3.1 Regularly Sampled Time Series (RSTS)

Consider a dataset represented by $D = \{X, Y\}$, where $X$ is a set of instances given by $X = \{X_1, ..., X_n\}$, $Y$ is the set of label given by $Y = \{y_1, ..., y_n\}$ and $n$ is the total number of instances. $X_i$ is a time series for $i^{\text{th}}$ instance given by $X_i = \{X_{i,1}, ..., X_{i,l_i}\}$ where $l_i$ is the number of time steps $i^{\text{th}}$ instance was measured. $X_{i,j}$ is the set of measured values of all sensors at time $t_{i,j}$, given by $X_{i,j} = \{x_{i,j}^1, ..., x_{i,j}^s\}$. Here, $x_{i,j}^m$ represents the measured value of sensor $m$ for $i^{th}$ instance at time $t_{i,j}$ and $s$ is the total number of sensors/features. We represent all sensors by their indices, and the set of indices of sensors is given by $\mathbb{M} = \{1, ..., s\}$. We present a snapshot of multi-variate RSTS data of $i^{\text{th}}$ instance in Fig. 2(a) considering $s = 3$ and $l_i = 4$. Note that $t_{i,2} - t_{i,1}$ is equal to $t_{i,3} - t_{i,2}$ in RSTS.

### 3.2 Irregularly Sampled Time Series (ISTS)

ISTS follows the definition of RSTS, except not all sensors will be measured at each time step, leading to an irregular sampling of each sensor, and $t_{i,2} - t_{i,1}$ need not be equal to $t_{i,3} - t_{i,2}$. ISTS is mathematically given as $X_{i,j} \subseteq \{x_{i,j}^1, ..., x_{i,j}^s\}$. A snapshot of ISTS for $i^{th}$ instance is shown in Fig. 2(b) where $X_{i,1} = \{x_{i,1}^1, x_{i,1}^3\}$, $X_{i,2} = \{x_{i,2}^2, x_{i,2}^3\}$, $X_{i,3} = \{x_{i,3}^1\}$ and $X_{i,4} = \{x_{i,4}^1, x_{i,4}^2\}$.

### 3.3 Problem Representation

For simplicity, we omit the subscript $i$ representing an instance and consider only one instance to discuss the problem and the working of the proposed model. In that sense, each measured value is given by $x_j^m$ (instead of $x_{i,j}^m$) and it is received at time $t_j$ (instead of $t_{i,j}$). Let us denote this instance by $Z = X_i$ and the set of values of measured sensors at time $t_j$ by $Z_j$. Based on this, at each time step $t_j$, we represent the measured sensors as $Z_j = (t_j, \bigcup_{\forall m \in \mathbb{A}_j} \{(x_j^m, \Delta_j^m)\})$ where $\mathbb{A}_j$ is the set of sensors measured at time $t_j$, given by $\mathbb{A}_j \subseteq \mathbb{M}$. $\Delta_j^m$ denotes the time delay between two successive values measured by sensor $m$, *i.e.*, $\Delta_j^m = t_j - t_k$, where $k = max(1, ..., j-1)$ such that $m \in \mathbb{A}_k$. Thus, the whole input data is given by $Z = \bigcup_{j=1}^{l} \{Z_j\}$ where $l$ is the number of time steps. Following the definition of ISTS, we visually present the data representation and the corresponding equations in Fig. 2(c).

## 4 SLAN: Switch LSTM Aggregation Network

The motivation of SLAN is propelled by the effectiveness of the sequence model, like LSTM, in handling time series data (Che et al., 2016). However, a single LSTM is incapable of modeling ISTS without imputation. Therefore, we devise a strategy of employing one LSTM per sensor. Since ISTS has irregular sampling, we propose a simple switch layer that facilitates the activation of only those LSTMs whose corresponding sensors are measured. Furthermore, we introduce global and local summary states to share information between all sensors.

SLAN is an adaptive LSTM-based model that dynamically changes its architecture depending on the measured sensors at any time point by utilizing a switch layer. The architecture of SLAN is presented in Fig. 3(a). It consists of a pack of LSTMs such that there is a one-on-one connection between a sensor and an

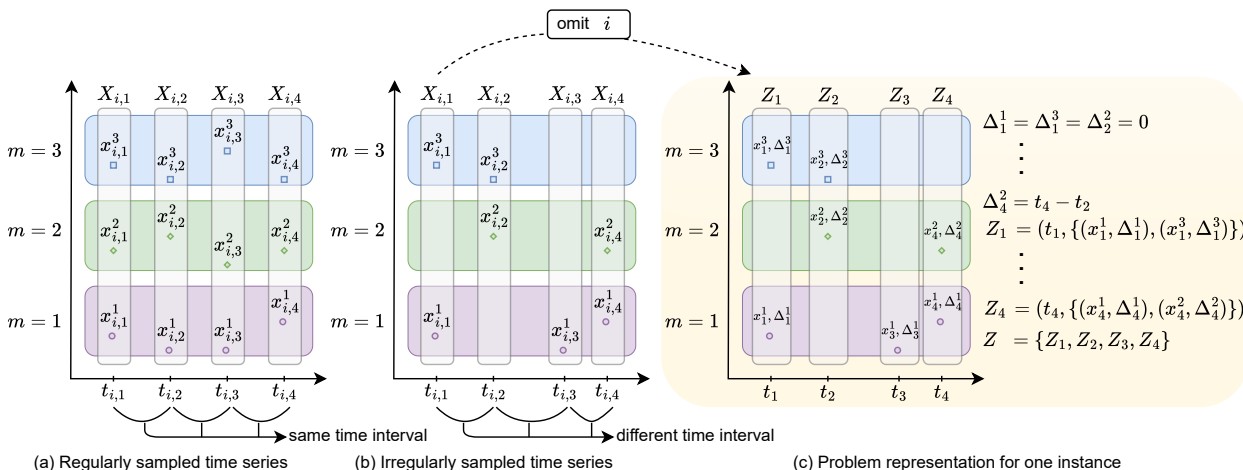

Figure 2: (a) A snapshot of multi-variate regularly sampled time series for $i^{th}$ instance. $m$ represents the index of the sensor. (b) A snapshot of multi-variate irregularly sampled time series (ISTS) for $i^{th}$ instance. (c) Problem representation of the ISTS with respect to one instance by omitting the subscript $i$. (Best viewed in color)

LSTM block. The switch layer facilitates this (see the yellow-colored box in Fig. 3(a)). Each sensor is connected to its corresponding LSTM block by a switch. A switch goes "on" if its corresponding sensor is measured; otherwise, it stays off. The "on" switch results in activating its corresponding LSTM block, thus, eliminating the need for any imputation. The LSTM block outputs a long-term memory (LTM) and a short-term memory (STM) (Fig. 3(b)). The LTM of each activated LSTM block is aggregated to produce a global summary state and passed on to all the LSTM blocks for the next time step as input. This aids the LSTM blocks with summarised information.

LSTM allows sequential processing of the time series, preserving their arrival order. However, still, it is necessary to model the time information associated with each input. This is more so in the case of ISTS since the time interval is not fixed. We draw upon the many methods presented in the literature to model time information and utilize Time2Vec (Kazemi et al., 2019) for its demonstrated effectiveness.

The previous short-term memory (STM) of each activated LSTM block is decayed based on the vector representation of time delay and decay function (discussed below). This decayed STM is passed as an input for the next measured time point. This acts as a local summary for each sensor. Finally, at the last time point, the STM(s) from each LSTM block is concatenated with the aggregated LTM as seen in the concat layer in Fig. 3(a). The concat layer is then fully connected to a 2-node output layer for binary classification. The fully connected layer can be easily extended for multi-class classification.

## 4.1 Architecture

SLAN consists of $s$ LSTM blocks $\{L^1, ..., L^s\}$ where $L^m$ is associated with sensor $m$. We define the switch layer ($\mathbb{S}_j$) as the set of switches kept "on" based on the measured sensors at time $t_j$. Since there is a one-on-one correspondence between a switch and its corresponding measured sensor, we borrow the representation of $\mathbb{S}_j$ as the indices of the sensors measured at time $t_j$ from section 3, thus $\mathbb{S}_j = \mathbb{A}_j$.

Each active LSTM block ($L^m$) at time $t_j$ takes the sensor value ($x_j^m$), STM ($h_{j-1}^m$), LTM ($c_{j-1}^m$) and time delay ($\Delta_j^m$) as inputs and outputs $h_j^m$ and $c_j^m$, given by

$$(h_j^m, c_j^m) = L^m(x_j^m, h_{j-1}^m, c_{j-1}^m, \Delta_j^m) \quad \forall m \in \mathbb{S}_j \tag{1}$$

where $L^m \; \forall m \in \mathbb{A}_j$ are active based on $\mathbb{S}_j$ at time $t_j$. An aggregate function is employed on the LTM of the active LSTM blocks to get a summary state ($c_j$) at $t_j$. Any function that can group multiple values to give a single summary value can be used as an aggregation function and is represented by $agg()$. Some examples

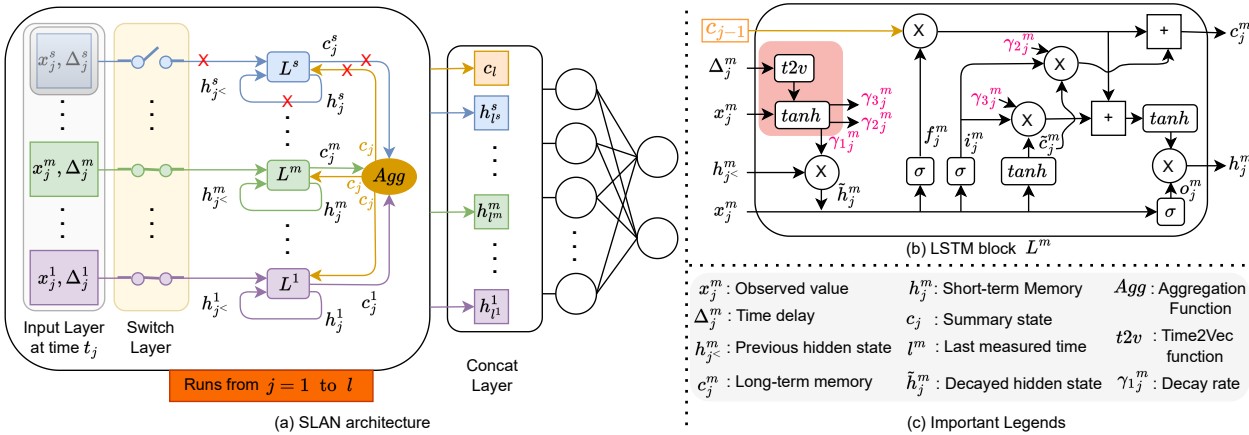

Figure 3: (a) SLAN Architecture. Here $x_j^m$ denotes the input at time $t_j$ of $m^{\text{th}}$ sensor. The closed circuit in the switch layer means that a particular switch is "on", otherwise it is "off". The X sign in red implies that there is no input or output to the corresponding LSTM block. (b) The inner working of an LSTM block is given here. (c) Notations are denoted in the legend.

of aggregation functions are mean, max, and attention. The summary state is given as

$$c_j = agg(\bigcup_{\forall m \in \mathbb{S}_j} \{c_j^m\}) \tag{2}$$

The $c_j$ is used as an input for the next time step for every active LSTM block. An active LSTM at $t_j$ might not be active at $t_{j-1}$. Thus, the STM input to $L^m$ at $t_j$ is represented by $h_{j<}^m$ (instead of $h_{j-1}^m$) where $j^< = \max(1, ..., j-1)$ such that $m \in \mathbb{S}_j$. Therefore, equation 1 can be updated as:

$$(h_j^m, c_j^m) = L^m(x_j^m, h_{j<}^m, c_{j-1}, \Delta_j^m) \quad \forall m \in \mathbb{S}_j \tag{3}$$

Finally, the hidden states (or STM) of all the LSTM blocks and the summary state are concatenated to give a final output. Note that all the sensors may not be observed in the last timestamp $t_l$. Therefore, we represent the last measure time for each sensor by $t_{l^m}$, such that $t_{l^m} \leq t_l$. Thus, the concat layer is given by $C = \{c_l, h_{l^1}^1, ..., h_{l^s}^s\}$. A fully connected network is employed to get a final prediction from $C$ as follows

$$\hat{y} = F(C) \tag{4}$$

**Hidden State Decay** Since the measurement of each sensor is irregular, we employ a time-decay function on the hidden states inspired by (Kazemi et al., 2019). The time-decay function ensures that the previous local summary is adjusted based on the time delay ($\Delta_j^m$) of each sensor. Since each sensor is different, the decaying function should differ for each sensor. Thus, a trainable time-decay function is employed. The decay function is given as

$$\gamma_{1j}^m = tanh(W_{\gamma_1}^m x_j^m + V_{\gamma_1}^m t2v(\Delta_j^m) + b_{\gamma_1}^m), \quad \text{where}$$
$$t2v(\Delta_j^m) = sin(\omega_j^m \Delta_j^m + \varphi_j^m) \tag{5}$$

Here, $t2v$ is the Time2Vec function with $\omega_j^m$, and $\varphi_j^m$ as the learnable parameters. The sine function in Time2Vec helps capture periodic behaviors without the need for feature engineering. The $W_{\gamma_1}^m, V_{\gamma_1}^m$, and $b_{\gamma_1}^m$ are the parameters of the decay function. Consequently, the decay of the hidden state is given by

$$\tilde{h}_j^m = \gamma_{1j}^m \odot h_{j<}^m \tag{6}$$

where $\odot$ is the element-wise dot product.

---

**Algorithm 1** Switch LSTM Aggregate Network

---

**Require:** Model $M$ with $s$ LSTM block, switch layer S, and aggregation function as shown in Figure 2(a)

**repeat**
   **for** $j$ in timestamp **do**
      Create switch layer $\mathbb{S}_j$ from measured sensors $\mathbb{A}_j$
      Activate LSTM blocks based on $\mathbb{S}_j$
      Calculate $h_j[\mathbb{S}_j]$, $c_j[\mathbb{S}_j]$ by equation 3
      Calculate $c_j$ using aggregation function by equation 2
   **end for**
   Concat all final hidden states ($h_{l^m}^m$) and final summary state ($c_l$) to get concat layer $C$
   Predict $\hat{y}$ using equation 4
   Update $M$ based on the loss
**until** Batch Left to run

---

**Working of an LSTM block** We employ TimeLSTM (Zhu et al., 2017) as an LSTM block in our study. The gates of $L^m$ at time $t_j$ are denoted by forget gate ($f_j^m$), input gate ($i_j^m$), output gate ($o_j^m$) and cell state ($\tilde{c}_j^m$). Based on the decayed hidden states ($\tilde{h}_j^m$) given by equation 6 and summary state ($c_{j-1}$) given by equation 2, the gates are determined as

$$
\begin{aligned}
f_j^m &= \sigma(W_f^m x_j^m + V_f^m \tilde{h}_j^m + b_f^m) \\
i_j^m &= \sigma(W_i^m x_j^m + V_i^m \tilde{h}_j^m + b_i^m) \\
o_j^m &= \sigma(W_o^m x_j^m + V_o^m \tilde{h}_j^m + b_o^m) \\
\tilde{c}_j^m &= tanh(W_c^m x_j^m + V_c^m \tilde{h}_j^m + b_c^m)
\end{aligned}
\tag{7}
$$

The final short-term and long-term memory depends on the decayed cell state achieved via $\gamma_{2j}^m$ and $\gamma_{3j}^m$ (equation 5) and is given by

$$
\begin{aligned}
c_j^m &= f_j^m \odot c_{j-1} + i_j^m \odot \tilde{c}_j^m \odot \gamma_{2j}^m \\
h_j^m &= o_j^m \odot tanh(f_j^m \odot c_{j-1} + i_j^m \odot \tilde{c}_j^m \odot \gamma_{3j}^m)
\end{aligned}
\tag{8}
$$

**Algorithm** The pseudo-code of SLAN is presented in the Algorithm 1.

## 4.2 Unrolled SLAN

An unrolled architecture based on the snapshot of an instance presented in Figure 2(c) is shown in Figure 4 (from left to right). We present the detailed workflow of the unrolled SLAN architecture here. The progression of SLAN at each time step is discussed next.

**Time $t_1$** We receive input $Z_1 = (t_1, \{(x_1^1, \Delta_1^1), (x_1^3, \Delta_1^3)\})$. Based on the measured sensors, the switch layer is $\mathbb{S}_1 = \{1, 3\}$, indicating switch 1 and switch 3 are "on". Thus, the associated LSTM blocks $L^1$ and $L^3$ are activated. The hidden states ($h_0^1, h_0^2, h_0^3$) and summary state ($c_0$) is initialized randomly. The time delay is $\Delta_1^1 = \Delta_1^3 = 0$. Using equation 3, we get $(h_1^1, c_1^1) = L^1(x_1^1, h_0^1, c_0, \Delta_1^1)$ and $(h_1^3, c_1^3) = L^3(x_1^3, h_0^3, c_0, \Delta_1^3)$ where the inner working of $L^m$ is given by equation 6 and 7. The LTM ($c_1^1, c_1^3$) is aggregated to give the next summary state $c_1$.

**Time $t_2$** At $t_2$, $Z_2 = (t_2, \{(x_2^2, \Delta_2^2), (x_2^3, \Delta_2^3)\})$, thus $\mathbb{S}_2 = \{2, 3\}$. Corresponding LSTM $L^2$ and $L^3$ are kept active. The previous hidden state of $L^2$ and $L^3$ are $h_0^2$ and $h_1^3$ respectively. The time delay is $\Delta_2^2 = \Delta_2^3 = t_2 - t_1$. We calculate $(h_2^2, c_2^2) = L^2(x_2^2, h_0^2, c_1, \Delta_2^2)$ and $(h_2^3, c_2^3) = L^3(x_2^3, h_1^3, c_1, \Delta_2^3)$ using equation 3. Based on this, the summary state $c_2$ is given by $agg(c_2^2, c_2^3)$.

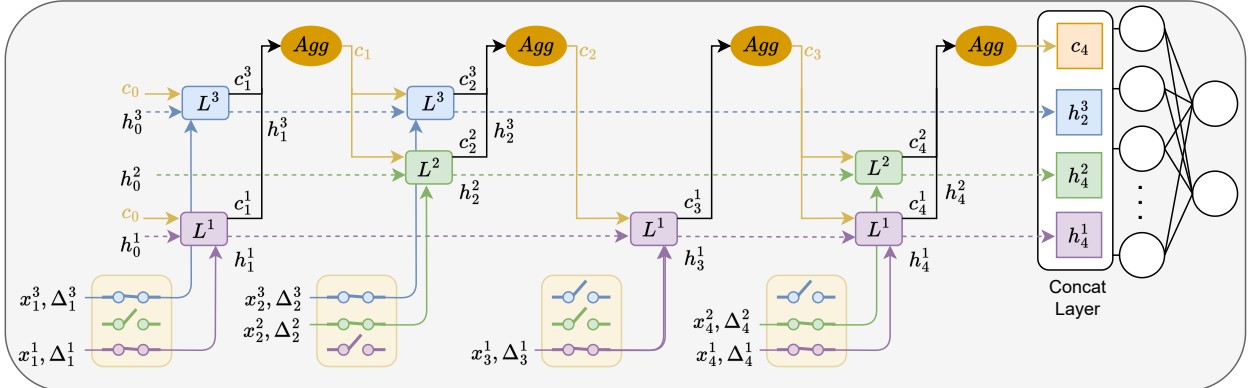

Figure 4: Unrolled SLAN architecture based on the example in Figure 2(c). (Best viewed in color)

**Time** $t_3$    The input is $Z_3 = (t_3, \{(x_3^1, \Delta_3^1)\})$, time delay is $\Delta_3^1 = t_3 - t_1$ and switch layer is $\mathbb{S}_3 = 1$ . Thus we compute $(h_3^1, c_3^1) = L^1(x_3^1, h_0^1, c_2, \Delta_3^1)$ and $c_3 = agg(c_3^1)$.

**Time** $t_4$    We get $(h_4^1, c_4^1) = L^1(x_4^1, h_3^1, c_3, \Delta_4^1)$ and $(h_4^2, c_4^2) = L^2(x_4^2, h_2^2, c_3, \Delta_4^2)$ where $\Delta_4^1 = t_4 - t_3$ and $\Delta_4^2 = t_4 - t_2$. Finally, the hidden states $(h_4^1, h_4^2, h_2^3)$ and the summary state $c_4$, where $c_4 = agg(c_4^1, c_4^2)$, are concatenated to give $C = \{c_4, h_4^1, h_4^2, h_2^3\}$. A fully connected layer is employed to give the final prediction as $\hat{y} = F(C)$ (see equation 4). This demonstrates the simplicity of SLAN in dynamically adapting the irregularly sampled sensor measurements without any need for imputation.

### 4.3   Discussion

**SLAN vs TimeLSTM and Time2Vec**   It is important to note that neither TimeLSTM nor Time2Vec is equipped to manage missing data in ISTS datasets. In contrast, SLAN effectively addresses this issue by employing a group of TimeLSTM units (enhanced by Time2Vec for decay functions), a simple switch strategy, and the sharing of information between TimeLSTM units through both global and local summary states. This framework allows SLAN to model ISTS data without the need for imputation.

**Switch Layer in SLAN vs Observation Mask in Transformer**   Note that the switch layer in SLAN is different from the observation mask in Transformers (Vaswani et al., 2017). The switch layer dynamically changes the architecture of SLAN to adapt to missing values by explicitly informing the model which LSTM blocks will be active. The architecture of the Transformer is fixed, where the observation masks are concatenated with the input value and passed as input to the model. The Transformer implicitly learns the meaning of observational masks via training.

**SLAN vs GRU-D**   When data is missing, GRU-D performs input imputation and hidden state decay. The input is imputed as $x_t^d = m_t^d x_t^d + (1 - m_t^d)\gamma_{x_t^d} x_{t'}^d + (1 - m_t^d)(1 - \gamma_{x_t^d})\tilde{x}^d$ where $m_t^d$ is the masking value, $\gamma$ is the decay factor, $x_{t'}^d$ is the last observation of the $d^{\text{th}}$ variable ($t^{'} < t$) and $\tilde{x}^d$ is the empirical mean of the $d^{\text{th}}$ variable. The hidden state is decayed as $\gamma_j = exp\{-max(0, a_\gamma \Delta_j + b_\gamma)\}$. When the data is not missing, GRU-D just performs the hidden state decay to capture richer knowledge from missingness. Thus, *GRU-D performs imputation when the data is missing.* Whereas, in SLAN, when data is missing, the switch of the LSTM corresponding to that data value is 'off'. Hence, *SLAN don't perform any form of imputation.* When data is not missing, SLAN performs the hidden state decay (equation 5 and 6) using Time2Vec to capture information from time delay ($\Delta_j^m$). Hence, SLAN differs from GRU-D, and unlike GRU-D, SLAN is a non-imputation model.

# 5 Experiments

## 5.1 Datasets

We consider MIMIC-III (M-3) (Johnson et al., 2016) and Physionet 2012 (P-12) (Goldberger et al., 2000) datasets to showcase the efficacy of SLAN. We prepare the datasets by following SeFT (Horn et al., 2020). For both datasets, the mortality prediction task is considered. The datasets are skewed with 13.22% and 14.24% positive labels for M-3 and P-12, respectively, making them challenging datasets. The number of missingness is $1.8 \times 10^7$ and $2.8 \times 10^7$ for the M-3 and P-12 datasets, leading to high irregularity. A detailed description of the dataset is presented in Table 1. It is important to note that several studies in the literature have also utilized the Physionet 2019 (P-19) dataset. However, due to its substantial size and resource constraint, we could not benchmark on the P-19 dataset.

Table 1: Dataset Description. #Instances is the number of patient records in the datasets, #Sensors is the number of features/sensors in each instance, # Static is the number of static variables, #Observations is the average number of observations recorded in each instance, i.e., the number of time steps, #Num-Imputation is the number of imputation or missing values and Imbalance is the percentage of instances with a minority class label.

| Dataset | MIMIC-III | Physionet 2012 |
|---|---|---|
| #Instances | 22110 | 11988 |
| #Sensors | 17 | 37 |
| #Static | 0 | 6 |
| #Observations(avg.) | 77.7 | 74.9 |
| #Num-Imputation | $1.8 \times 10^7$ | $2.8 \times 10^7$ |
| Imbalance (%) | 13.22 | 14.24 |

**MIMIC-III**   It is a dataset of stays of patients in the critical care unit at a large tertiary care hospital. It has 21142 stays of unique patients (instances) with a median length of stay of 2.1 days. A total of 17 physiological measurements, like vital signs, medications, etc., are recorded for each patient. Following SeFT (Horn et al., 2020), we remove 32 instances. The discarded instances contained dramatically different recording frequencies compared to the rest of the dataset. Thus, the total number of instances is 21110. We train our model for the in-hospital mortality prediction tasks. Some of the features with numerical data type have extreme outlier values, and hence are removed (see section B in Appendix).

**Physionet 2012**   It is a dataset of 12000 patient records (instances) containing measurements taken during the first 48 hours of the ICU stays. Each instance is associated with 37 time series variables (sensors) like blood pressure, lactate, respiration rate, etc., and 6 static descriptor features (i.e., RecordID, Age, Gender, height, ICUType, and Weight). We follow the SeFT (Horn et al., 2020) paper and remove 12 instances that do not contain any time series information. The weight feature is considered a time series since it is measured multiple times in the observation period. The final dataset has 11988 instances with 37 features. We train our model on the in-hospital mortality task, which is a binary classification task to predict if the patient dies before being discharged by using the data of the first 48 hours of the ICU admission.

## 5.2 Baselines

We consider both non-imputation and imputation baselines. Among imputation, GRU-D (Che et al., 2016), IP-Nets (Shukla & Marlin, 2018), and ViTST (Li et al., 2024) are considered. The non-imputation baselines are Transformer (Vaswani et al., 2017), SeFT (Horn et al., 2020), Raindrop (Zhang et al., 2021), CoFormer (Wei et al., 2023), and IVP-VAE (Xiao et al., 2024). See section C in the Appendix for the implementation details of baseline models. Some recent models were excluded from our baselines either due to their implementation complexity or because they are forecasting models rather than classification models. More information about these models is provided in section D in the Appendix. For a fair comparison, we only included models capable of performing classification tasks in their original form.

Table 2: Comparison of various methods on M-3 and P-12 datasets. The **best** and 2nd best performance is represented by **bold** and underline, respectively. The metric is reported as the mean $\pm$ standard deviation of three runs with different seeds.

| Type | Model | MIMIC-III | | Physionet 2012 | |
|---|---|---|---|---|---|
| | | AUPRC | AUROC | AUPRC | AUROC |
| Imputation | GRU-D | $45.91 \pm 1.34$ | $83.43 \pm 0.66$ | $49.63 \pm 1.17$ | $84.94 \pm 0.29$ |
| | IPNets | $48.70 \pm 0.67$ | $84.90 \pm 0.26$ | $50.02 \pm 0.61$ | $85.54 \pm 0.42$ |
| | ViTST | $47.88 \pm 0.49$ | $85.49 \pm 0.82$ | $48.53 \pm 1.05$ | $84.27 \pm 0.37$ |
| Non-Imputation | Transformer | $48.88 \pm 1.01$ | $84.89 \pm 0.53$ | $49.37 \pm 0.77$ | $84.23 \pm 0.14$ |
| | SeFT | $46.01 \pm 1.06$ | $85.43 \pm 0.26$ | $50.69 \pm 0.89$ | $85.28 \pm 0.28$ |
| | Raindrop | $35.76 \pm 0.29$ | $77.18 \pm 0.20$ | $42.28 \pm 1.48$ | $79.34 \pm 0.19$ |
| | CoFormer | $50.51 \pm 0.90$ | $85.08 \pm 0.56$ | $48.67 \pm 2.55$ | $85.12 \pm 0.96$ |
| | IVP-VAE | $47.02 \pm 0.75$ | $84.80 \pm 0.19$ | $47.35 \pm 0.72$ | $85.12 \pm 0.59$ |
| | **SLAN** | **$51.12 \pm 0.57$** | **$85.63 \pm 0.07$** | **$55.20 \pm 0.65$** | **$86.42 \pm 0.13$** |

## 5.3 Comparison Metrics

The datasets are imbalanced. Thus, we use the area under the receiver operating characteristic (AUROC) and the area under the precision-recall curve (AUPRC) as comparison metrics. AUROC informs the model's discriminative ability between positive and negative labels. Different true positive rates (TPR) and false positive rates (FPR) are achieved based on different thresholds for binary classification. This gives an ROC curve, and the area under this curve is AUROC. AUPRC is similar to AUROC, but instead of the TPR as the y-axis, precision is used, and instead of FPR as the x-axis, recall is used. It is mainly used for imbalanced data where the focus is on correctly classifying positive labels.

## 5.4 Implementation Details

We consider the train-val-test split of all datasets provided in SeFT. To handle the imbalance, we resort to a weighted oversampling strategy. Weighted oversampling involves preparing the training batch by sampling the data based on the class weights given by the inverse frequency of the class. The models are trained for 20 epochs with an early stopping of 5 on AUPRC to avoid overfitting. SLAN uses cross-entropy loss, AdamW optimizer, data standardization, and mean aggregate function. The size of short-term and long-term memory size is 64, and the learning rate is 0.0005. The learning rate is adaptive with decay by a factor of 0.5 after each epoch without improvement. The batch size is 16, and the dimension of the time embedding vector is 16 for both datasets. Since the P-12 dataset has 6 static features, the embedding of these features is concatenated in the final concat layer before applying a fully connected layer for prediction. The size of the embedding is kept equal to the size of the global summary state. We ran experiments on an NVIDIA DGX A100 machine. More details on implementation are given in section C in the Appendix.

## 5.5 Results

The performance of SLAN on M-3 and P-12 datasets are presented in Table 2. SLAN performs competitively on both datasets across both metrics. SLAN outperforms the second-best results by 1.2% and 8.9% in absolute AUPRC points, and 0.2% and 1% in absolute AUROC points for M-3 and P-12, respectively.

# 6 Ablation Studies

Unless otherwise stated, all the below experiments of SLAN are performed by following the implementation details given in the previous section.

**Aggregation Methods** We compare the performance of SLAN for mean, max, and simple attention (Bahdanau et al., 2014) as the aggregation function to calculate the global summary state. In attention, the normalized weightage of the LTM of each active LSTM block is determined using a single-layer feed-forward neural network, followed by the weighted average of LTMs to output the global summary state. In max, the element-wise max is performed over the candidate summary states. The comparison is reported in the middle part of Table 5. The performance of max is lower than both attention and mean because max may downplay the contribution of

Figure 5: Comparison of SLAN for different aggregation functions and variants of concat layer. Att stands for attention. G.S. stands for global summary state and L.S. stands for local summary state. G.S. + L.S. is the default setting of SLAN.

| | MIMIC-III | | Physionet 2012 | |
|---|---|---|---|---|
| | AUPRC | AUROC | AUPRC | AUROC |
| *Imputation* | | | | |
| ffill | **51.46±0.49** | 85.18±0.46 | 51.06±0.49 | 85.07±0.37 |
| mean | 48.73±0.79 | 84.30±0.36 | 51.65±0.73 | 85.28±0.32 |
| inter. | 49.44±0.26 | 84.96±0.31 | 50.75±0.07 | 84.88±0.34 |
| None | 51.12±0.57 | **85.63±0.07** | **55.20±0.65** | **86.42±0.13** |
| *Aggregation Function* | | | | |
| Max | 49.24±0.88 | 85.40±0.29 | 54.36±0.89 | 85.95±0.19 |
| Att | 50.38±0.96 | 85.59±0.47 | **55.37±0.10** | **86.44±0.16** |
| Mean | **51.12±0.57** | **85.63±0.07** | 55.20±0.65 | 86.42±0.13 |
| *Concat* | | | | |
| Only G.S. | 46.61±0.83 | 84.63±0.27 | 48.81±1.84 | 83.04±0.45 |
| Only L.S. | 50.96±0.51 | **85.80±0.42** | 54.43±0.31 | 86.20±0.20 |
| G.S.+L.S. | **51.12±0.57** | 85.63±0.07 | **55.20±0.65** | **86.42±0.13** |

many LTMs by highlighting just one, thus becoming sensitive to outliers. Among mean and attention, attention performs the best in P-12, whereas mean gives better results in M-3. Overall, the mean performs well since attention outperforms the mean by a margin of only 0.17 (AUPRC) and 0.02 (AUROC) in P-12, whereas it underperforms the mean by a margin of 0.74 (AUPRC) and 0.04 (AUROC) in M-3.

**Performance vs the best imputation model** We compare SLAN with the best imputation model to assess SLAN's robustness under an increased number of missing observations. IP-Nets perform the best among the imputation models, as evident from Table 2. IP-Nets surpasses other imputation models in both evaluated metrics on the P-12 and the AUPRC metric on the M-3 dataset. We randomly drop 25%, 50%, and 75% of observed data in both the M-3 and P-12 datasets. As demonstrated in Figure 1, SLAN consistently outperforms IP-Nets across all scenarios. SLAN achieves gains in absolute AUPRC points of 7.32%, 5.49%, and 6.85% in the M-3 dataset and 7.18%, 7.17%, and 12.63% in P-12 for the respective data drop of 25%, 50% and 75%. These results assert the superiority of SLAN even in conditions characterized by a substantial proportion of missing observations.

**Imputed SLAN** To determine the efficacy of SLAN, we compare it with imputed SLAN. We consider three types of imputation, namely, forward fill (ffill), mean, and interpolation imputed via the last measured value, global mean, and linear interpolation, respectively. As evident from Table 5, SLAN (represented by None) outperforms mean and interpolation. SLAN further surpasses ffill in the P-12 dataset and in the AUROC metric of the M-3 dataset.

**De-mistifying Concat Layer** SLAN's concat layer consists of a global summary state and the local summary state of each sensor. We remove the global summary state from the concat layer to check the informativeness of the local summary state (*Only L.S.* in Table 5). When compared with the default setting of SLAN, *i.e. G.S. + L.S.*, *Only L.S.* is slightly poorer (max by ∼1.39%) and even surpasses in AUROC on M-3 by 0.2%. This is because the local summary state contains individual sensor information and is also aided by the global summary state at each time step. Only global summary states in the concat layer (*Only G.S.*) perform 11.58% and 4.07% poorer than *G.S. + L.S.* in terms of AUPRC and AUROC on P-12. *Only G.S.* performs 8.82% and 1.17% poorer than *G.S. + L.S.* in AUPRC and AUROC, respectively for

M-3. Indeed *Only G.S.* performs significantly poorer than *G.S. + L.S.*, still it contains sufficient summarised information to outperform baseline models like Raindrop in both datasets and GRU-D in M-3 (Table 2).

**Data Scalability**  In the practical setting, it is important for any model to have data scalability, meaning the performance of the model on test data should improve as the amount of training data increases. We consider the first 25%, 50%, 75%, and 100% training data for both P-12 and M-3 and train our model on them. The average and the 95% confidence interval of 3 different runs of SLAN on the test data are shown in Figure 6. The performance of SLAN steadily increases with the increasing amount of training data on both datasets. The percentage improvement of AUPRC for M-3, when trained on 50% data compared to 25% data is 4.25%, 75% data compared to 50% data is 3.17%, and 100% data compared to 75% data is 1.43%. Transitioning from 25% to 50% data, we double the number of instances; thus, the percentage improvement is the highest. Whereas when trained on 100% data compared to 75% data, we add only 1/3rd data; thus, the percentage improvement is lowest. The same trend is followed in the AUROC of M-3, AUPRC, and AUROC of P-12. See section E in the Appendix for the exact metrics value.

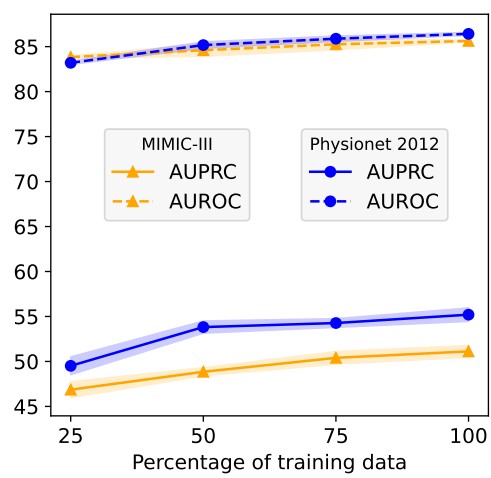

Figure 6: SLAN on different percentages of training datasets. The average with 95% confidence interval of 3 runs is reported here.

**Time Requirement and Scalability to Number of Sensors**  The worst-case time complexity of SLAN is given by $O((N/B)*T*K)$, where $N$ is the number of instances, $B$ is the batch size, $T$ is the maximum length of the time series, and $K$ is the time complexity to process a single LSTM. We utilize GPU to run SLAN, and therefore, the time complexity to process a single LSTM is $O(H^2)$, where $H$ is the hidden size of LSTM. For each LSTM, at each timestep, input, and weight tensors are constructed of shape $(F*B, H+1, 1)$ and $(F*B, H, H+1)$, which are then further multiplied using the torch.matmul operation, as weight∗input=output, with output shape of $(F*B, H, 1)$. Here, $F$ is the number of sensors. The first dimension given by $F*B$ is parallelized in GPU, so $F*B$ numbers of matrix multiplication are computed in parallel. The overall time complexity becomes equal to the time complexity to multiply a single matrix since all matrices are computed parallelly. Matrix multiplication of matrices with shape $(H, H+1)$ with $(H+1, 1)$ has a time complexity of $O(H^2)$. Since the factor $H^2$ comes from parallelized operation in GPUs, it will, therefore, have a very small constant factor compared to the other part $[(N/B)*T]$, which is processed sequentially. Therefore, the worst time complexity of SLAN in GPU is given by $O((N/B)*T*H^2)$. It can be seen that the training time of SLAN is dependent on the #Instances ($N$) and the #Observations ($T$). Refer to Table 1 for the definition of #Instances and #Observations. This is also evident in the training time required for M-3 and P-12. SLAN requires training time of 373.55±4.80 and 183.99±9.45 seconds per epoch (s/ep) for M-3 and P-12, respectively. Therefore, the time required for an instance of M-3 and P-12 is $2.55×10^{-2}$ and $2.40×10^{-2}$ s/ep, respectively. M-3 requires slightly more time than P-12 because its #Observations are slightly higher. Note that the time required by SLAN does not depend on the number of sensors, as M-3 has 17 sensors, whereas P-12 has 37. Thus, SLAN is scalable to the number of sensors with regard to time complexity.

**Space Complexity**  The space complexity of the SLAN can be given by $O(F*L + K + D)$, where $O(L)$, $O(K)$, and $O(D)$ are the space complexity of a single LSTM, final prediction network, and Time2Vec function, respectively. The three gates and cell state of an LSTM, given by equation 7, account for $4H^2+8H$ parameters. The three time decay function (see equation 5) requires $3H^2 + 6H$ parameters. Therefore, an LSTM requires $7H^2 + 14H$ parameters. The $O(K) = 2FH + 2H + 2$, and $O(D) = E$, where $E$ is the time embedding dimension. Therefore, the total number of learnable parameters in SLAN is $7FH^2 + 16FH + 2H + E + 2$. The values of $H$ and $E$ are 64 and 16, respectively, for both M-3 and P-12. Therefore, the total number of parameters for M-3 ($F = 17$) and P-12 ($F = 37$) is ∼504K and ∼1 million parameters, respectively. Furthermore, the number of parameters required for 1000 sensors would be ∼30 million, which

amounts to $\sim$0.22GB memory with 64-bit precision. Therefore, SLAN is scalable compared to large language models, such as GPT-like models, which require significant computing resources.

**Sampling Rate vs Importance of Sensors**   We use simple attention (Bahdanau et al., 2014) in the $Agg()$ unit to calculate the global summary state, which takes as input a set of LTMs $c_j^m$ from the activated $L^m$ at any time $t_j$. Our attention module contains a feed-forward neural network $\mathtt{nn}(.)$ which calculates a set of scores for each $c_j^m$ as $\mathrm{score}_j^m = \mathtt{nn}(c_j^m)$ and are used to obtain the attention weights as $a_j^m = \mathrm{score}_j^m / \sum_{k \in \mathbb{S}_j} \mathrm{score}_j^k$. We explore $a_j^m$ as an attempt to interpret the characteristics of the information encoded in the global summary state. This ablation is motivated by the empirical evidence for global summary (only G.S. in Table 5), which performs better than some of the previous baselines. The sampling rate denotes the number of measurements per hour of a particular sensor. Ideally, a sensor with a high sampling rate should hold high importance since the model sees it most often. We consider the M-3 dataset

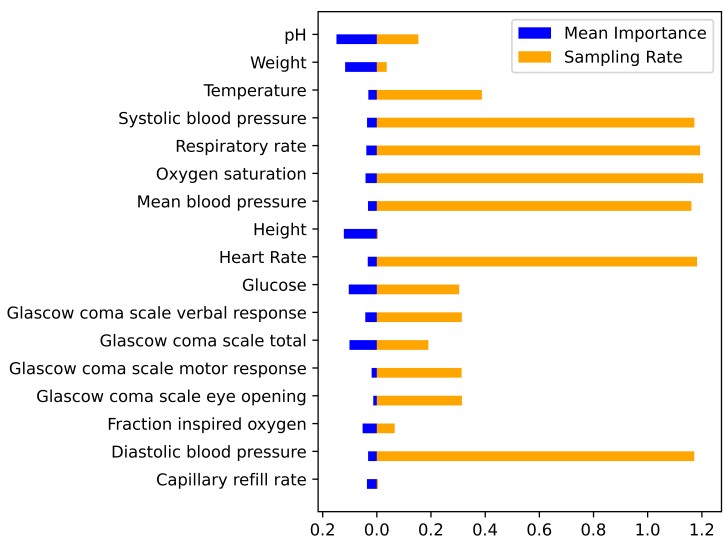

Figure 7: Comparison of the ranking of clinical variables *w.r.t.* sampling rate and mean importance.

for this ablation study. We sum the attention weights of each sensor across all the time steps, for each instance $i$. This is given by $\mathtt{sumI}^m = \sum_{i=1}^{n_{test}} \sum_j a_{i,j}^m$, such that $m \in \mathbb{S}_j$ for $i^{\text{th}}$ instance and $n_{test}$ is the number of instances in the test dataset. Here $a_{i,j}^m$ represents the attention weight of $m^{\text{th}}$ sensor at time $t_j$ for $i^{\text{th}}$ instance. The summation weights of each sensor are then divided by the number of times each sensor is measured in the test dataset (denoted by $C^m$) as $\mathtt{meanI}^m = \mathtt{sumI}^m / C^m$. This is then normalized to get the importance of each sensor as $\mathtt{normI}^m = \mathtt{meanI}^m / \sum_{k=1}^s \mathtt{meanI}^k$. We compare the mean importance ($\mathtt{normI}^m$) and the sampling rate ($r^m$) of sensors in Figure 7. Sensors with higher sampling rates are oxygen saturation, respiratory rate, and heart rate. Whereas the most important sensors are pH, height, and weight. Even though pH has the fifth-lowest sampling rate, it is the most important sensor in providing inference. Thus, frequently measured sensors may not be the most important sensors.

## 7   Conclusion

We propose a Switch LSTM Aggregate Network to handle multivariate ISTS data without any imputation. The optimal performance of SLAN and various ablation studies empirically demonstrate the effectiveness of our proposed model. We also establish the superiority of SLAN compared to the imputation model even when additional data is missing. Moreover, the SLAN framework can be extended for modeling multi-modality data, like adding clinical notes to sensor measurement data (Zhang et al., 2023). SLAN can be used for forecasting tasks by inheriting the idea of multi-horizon forecasting (Lim et al., 2021a). SLAN can also be leveraged for streaming data modeling in an online setting with time-variant dimensions (Agarwal et al., 2023). We plan to explore the above fields in the future.

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

## A  Imputation-Based Baseline Models

### A.1  GRU-D

Che et al. (2016) proposed GRU-D, which exploits missingness by considering two main missingness representation methods, masking, and timestamps, to devise effective solutions to characterize the missing patterns. The proposed model aims to use the masking information and temporal pattern in the missingness via the two trainable decay terms. The decay is calculated as

$$\gamma_t = exp\{-max(0, W_\gamma \delta_t + b_\gamma)\} \tag{9}$$

where $\gamma$ is the decay parameter at time $t$, $W$ and $b$ are model parameters to learn the decay. GRU-D decays the hidden states as

$$h_{t-1} = \gamma_{h_t} \odot h_{t-1} \tag{10}$$

where $h_{t-1}$ is the hidden state from time $t-1$ and $\gamma_{h_t}$ is decay value of hidden state at time $t$. GRU-D further imputes the input missing value whenever the input data is missing. The following equation does the imputation

$$x_t^d = m_t^d x_t^d + (1 - m_t^d)\gamma_{x_t^d} x_{t'}^d + (1 - m_t^d)(1 - \gamma_{x_t^d})\tilde{x}^d \tag{11}$$

Here, $m_t^d$ represents the masking value which is 1 if the sensor is measured otherwise 0, $\gamma_{x_t^d}$ is the decay factor, $x_{t'}^d$ is the last observation of the $d^{\text{th}}$ variable ($t' < t$) and $\tilde{x}^d$ is the empirical mean of the $d^{\text{th}}$ variable. Thus, the missing input feature is imputed whenever not measured.

### A.2  IP-Nets

Shukla & Marlin (2018) proposed Interpolation-Prediction Networks, which consist of an interpolation network followed by a prediction network. IP-Nets convert ISTS to regularly sampled time series (RSTS) in the interpolation network. It uses the information from each time series to interpolate values of all the other time series. IP-Nets considers a set of reference time points $r = [r_1, ..., r_T]$. All the reference time points are evenly spaced within its interval. For each sensor of an instance, IP-Nets output three interpolants (cross-channels, transient component, and intensity) corresponding to each reference point and a sensor. Thus, the interpolation network takes $i^{\text{th}}$ ISTS instance ($X_i$) as input and outputs $i^{\text{th}}$ RSTS interpolated output ($\hat{X}_i$) where the dimension of $\hat{X}_i$ is $(3s) \times T$. Here, $s$ is the number of sensors/features, $T$ is the number of reference time points, and 3 represents the number of interpolants corresponding to each time point for each sensor. Finally, in the prediction network, $\hat{X}_i$ is used as an input to produce the final prediction as $\hat{y}_i = g_\theta(\hat{X}_i)$.

### A.3  ViTST

Li et al. (2024) introduced the Vision Time Series Transformer, which converts each sample of ISTS data into line graphs. These graphs are subsequently organized into a standard RGB image format. The process involves plotting timestamps on the horizontal axis and observed values on the vertical axis of the line graph, with observations connected chronologically using linear interpolation to address missing values. Each sensor or feature generates a line graph that is arranged into a single image following a predefined layout. The vision transformer, specifically the Swin Transformer, is utilized for the classification of the created image. To integrate static features, ViTST transforms them into text using a template and encodes this text with a RoBERTa-base text encoder. The text and image embeddings are then concatenated to facilitate classification.

## B  MIMIC - Outlier Removal

Some of the features with numerical data type have extreme outlier values, like oxygen saturation, which should have values in the range of 0-100, but some values are in the range of $10^5$ (see Figure 8a), possibly due to input/formatting error. Therefore, we remove these outliers. From the training data, 0.008% extreme values are removed in each numerical feature. 0.008% is selected based on the histogram chart of each feature

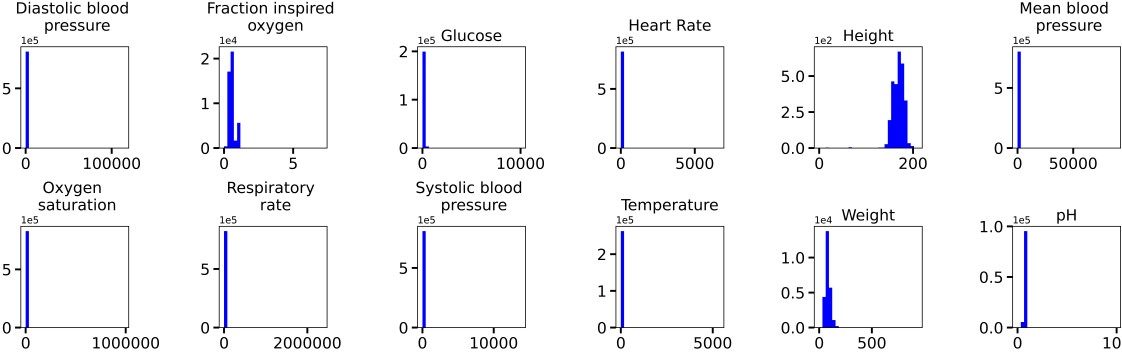

(a) Histogram of the numerical features of the MIMIC-III dataset before outlier removal.

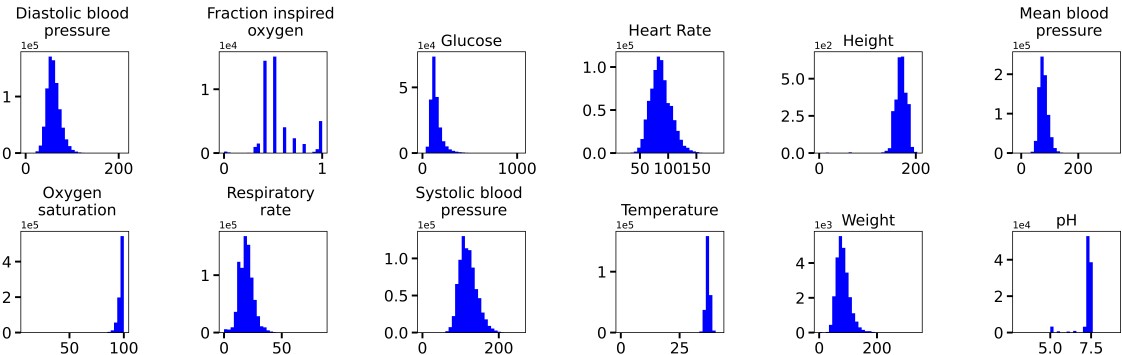

(b) Histogram of the numerical features of the MIMIC-III dataset after outlier removal.

Figure 8: Change in the distribution of numerical features in MIMIC-III dataset after removing 0.0008% extreme outlier values.

in the training data, as it does not cause too much loss of information and forms a well-distributed histogram, as shown in Figure 8b. Based on the lower and upper bound values with respect to 0.008% extreme values, the outliers from the test and validation data are also removed.

# C   Implementation Details

We have three splits of data: train, validation, and test, as mentioned in the SeFT (Horn et al., 2020) article. The best value of the hyperparameter for the models is determined on the validation set and subsequently used on the test set to evaluate the final performance. Hyperparameter searching is conducted sequentially for each parameter, as detailed in section C.1. Additional settings such as early stopping, learning rate decay, standardization, weighted oversampling, and the number of epochs are heuristically set and uniformly applied across all models, unless specified otherwise. These settings are defined in the section 5.4 of the main manuscript.

All the experiments are run on an NVIDIA DGX A100 machine equipped with 8 40 GB GPUs. Each model is executed three times using random seeds of 2024, 2025, and 2026 to ensure reproducibility. The range of hyperparameter values searched, and its best value for all the models on each dataset is documented in Table 3.

For the P-12 dataset, which includes static features, an embedding is generated using a linear layer. This embedding is concatenated to the final embedding of the respective method before proceeding with a feed-forward neural network for classification. This procedure is consistently applied across all models unless specified otherwise.

Table 3: All the hyperparameters used in each model, their search values, and the best value of each hyperparameter. ViTST requires substantial running time and resources. Therefore, we resort to the best hyperparameters reported in the ViTST paper.

| Model | Hyperparameter(s) | Search | Best Values M-3 | Best Values P-12 |
|---|---|---|---|---|
| GRU-D | Hidden Size | {16, 32, 64, 128, 256} | 256 | 16 |
| | Batch Size | {16, 32, 64, 128, 256, 512} | 16 | 16 |
| | Learning Rate | {5e5, 1e4, 5e4, 1e3, 5e3, 1e2, 5e2} | 5e3 | 1e3 |
| | Dropout | {0, 0.1, 0.2, 0.3, 0.4} | 0.1 | 0.1 |
| | Recurrent Dropout | {0, 0.1, 0.2, 0.3, 0.4} | 0 | 0.5 |
| IPNets | Hidden Size | {16, 32, 64, 128, 256} | 128 | 128 |
| | Batch Size | {16, 32, 64, 128, 256, 512} | 16 | 32 |
| | Learning Rate | {5e5, 1e4, 5e4, 1e3, 5e3, 1e2, 5e2} | 1e3 | 1e3 |
| | Imputation Step | {0.5, 1, 2.5, 5} | 2.5 | 5 |
| | Reconstruction Fraction | {0.05, 0.1, 0.2, 0.5, 0.75} | 0.2 | 0.2 |
| | Reconstruction Weights | {0, 0.5, 1, 1.5, 2} | 1.5 | 2 |
| | Dropout | {0, 0.1, 0.2, 0.3, 0.4} | 0.2 | 0.2 |
| | Recurrent Dropout | {0, 0.1, 0.2, 0.3, 0.4} | 0.1 | 0.3 |
| ViTST[*] | Batch Size | - | 48 | 48 |
| | Learning Rate | - | 2e5 | 2e5 |
| Transformer | Hidden Size | {16, 32, 64, 128, 256} | 32 | 64 |
| | Batch Size | {16, 32, 64, 128, 256, 512} | 64 | 16 |
| | Learning Rate | {5e5, 1e4, 5e4, 1e3, 5e3, 1e2, 5e2} | 5e4 | 5e4 |
| | Number of Layers | {1, 2, 3, 4} | 2 | 2 |
| | Number of Attention Heads | {2, 4, 8, 16} | 16 | 2 |
| | Maximum Timescale | {10, 100, 1000} | 100 | 100 |
| | Dropout | {0, 0.1, 0.2, 0.3, 0.4} | 0.1 | 0.2 |
| | Aggregation Function | {sum, max, mean} | mean | mean |
| SeFT | Batch Size | {16, 32, 64, 128, 256, 512} | 64 | 16 |
| | Learning Rate | {5e5, 1e4, 5e4, 1e3, 5e3, 1e2, 5e2} | 5e4 | 5e4 |
| | Number of Phi Layers | {1, 2, 3, 4, 5} | 1 | 1 |
| | Number of Psi Layers | {1, 2, 3, 4, 5} | 3 | 3 |
| | Number of Rho Layers | {1, 2, 3, 4, 5} | 1 | 2 |
| | Phi Width | {16, 32, 64, 128, 256, 512} | 64 | 64 |
| | Psi Width | {16, 32, 64, 128, 256, 512} | 64 | 16 |
| | Rho Width | {16, 32, 64, 128, 256, 512} | 512 | 512 |
| | Latent Width | {32, 64, 128, 256, 512, 1024, 2048} | 256 | 2048 |
| | Psi Latent Width | {32, 64, 128, 256, 512, 1024, 2048} | 128 | 64 |
| | Dot Product Dimension | {32, 64, 128, 256, 512, 1024, 2048} | 2048 | 128 |
| | Number of Attention Heads | {2, 4, 8, 16} | 4 | 16 |
| | Number of Positional Dimension | {4, 8, 16} | 8 | 8 |
| | Maximum Timescale | {10, 100, 1000} | 1000 | 100 |
| | Attention Dropout | {0, 0.1, 0.2, 0.3, 0.4} | 0 | 0.1 |
| | Phi Dropout | {0, 0.1, 0.2, 0.3, 0.4} | 0 | 0 |
| | Rho Dropout | {0, 0.1, 0.2, 0.3, 0.4} | 0.1 | 0.1 |
| Raindrop | Batch Size | {16, 32, 64, 128, 256, 512} | 128 | 32 |
| | Learning Rate | {5e5, 1e4, 5e4, 1e3, 5e3, 1e2, 5e2} | 5e4 | 5e4 |
| | Observation Embedding Size | {2, 4, 8, 16} | 16 | 8 |
| | Number of Layers | {1, 2, 3, 4} | 1 | 2 |
| | Number of Heads | {2, 4, 8, 16} | 2 | 2 |
| | Dropout | {0, 0.1, 0.2, 0.3, 0.4} | 0.3 | 0.3 |
| CoFormer | Batch Size | - | 16 | 16 |
| | Learning Rate | {5e5, 1e4, 5e4, 1e3, 5e3, 1e2, 5e2} | 1e4 | 5e4 |
| | Number of Layers | {2, 4, 6, 8} | 2 | 2 |
| | Number of Heads | {2, 4, 8, 16} | 8 | 2 |
| | Hidden Size | {16, 32, 64, 128, 256} | 256 | 128 |
| | Variate Code Dimension | {16, 32, 64, 128, 256} | 32 | 128 |
| | Dropout | {0, 0.1, 0.2, 0.3, 0.4} | 0.3 | 0.3 |
| IVP-VAE | Batch Size | {16, 32, 64, 128, 256, 512} | 64 | 32 |
| | Learning Rate | {5e5, 1e4, 5e4, 1e3, 5e3, 1e2, 5e2} | 1e3 | 5e3 |
| | Number of Layers | {1, 2, 3, 4, 5} | 2 | 5 |
| | Hidden Size | {16, 32, 64, 128, 256} | 128 | 32 |
| SLAN | Hidden Size | {16, 32, 64, 128, 256} | 64 | 64 |
| | Batch Size | {16, 32, 64, 128, 256, 512} | 16 | 16 |
| | Time Embedding Dimension | {16, 32, 64, 128, 256} | 16 | 16 |
| | Learning Rate | {5e5, 1e4, 5e4, 1e3, 5e3, 1e2, 5e2} | 5e4 | 5e4 |

### C.1 SLAN

The hyperparameters in SLAN include hidden size (dimensions of short-term and long-term memory), batch size, time embedding dimension, and learning rate. We opted for finding the best hyperparameter value on the validation set one at a time as follows:

1. Initially, we fixed the batch size to 32, the learning rate to 0.0005, the time embedding dimension to 16, and varied the hidden size to 16, 32, 64, 128, and 256.

2. Next, we varied the batch size to 16, 32, 64, 128, 256, and 512. Here, the learning rate is fixed to 0.0005, the time embedding dimension to 16, and the hidden size to the best value found in the previous step.

3. Based on the best-hidden size and batch size, we varied the time embedding dimension to 16, 32, 64, 128, and 256 with a fixed learning rate of 0.0005.

4. Finally, we vary the learning rate to 0.00005, 0.0001, 0.0005, 0.001, 0.005, 0.01, and 0.05.

The best value of all the hyperparameters is provided in Table 3.

### C.2 GRU-D, IPNets, Transformer, SeFT, and IVP-VAE

Similar to SLAN, all the hyperparameters of GRU-D (Che et al., 2016), IPNets (Shukla & Marlin, 2018), Transformer (Vaswani et al., 2017), SeFT (Horn et al., 2020), and IVP-VAE (Xiao et al., 2024) are determined sequentially in the order mentioned in Table 3. The best values of hyperparameters are also documented in Table 3.

### C.3 ViTST

We implemented ViTST (Li et al., 2024) by adhering to the methodologies described in the ViTST article and its accompanying code[1]. ViTST necessitates a predefined grid layout to generate images for each instance. Specifically, a 4×5 grid layout is used for the M-3 dataset, while a 6×6 grid layout is employed for the P-12 dataset. It is important to note that the P-12 dataset comprises 37 features, yet the grid layout accommodates only 36 features. Following the original paper's guidelines, we observed that one of the feature values consistently equals 1. Therefore, we stick to a 6×6 grid layout. Each grid cell measures 64×64, resulting in total image dimensions of 256×320 for the M-3 dataset and 384×384 for the P-12 dataset. Linear interpolation is utilized to impute missing values. To create the image, the line style is set to '-' with a line width of 1, and observed values are indicated by '*' with a marker size of 2. Rather than employing weighted oversampling, we adopted the sampling technique from the original paper, which equalizes the number of samples across classes by matching the count of minority class samples to that of the majority class samples. Given the substantial time requirement of ViTST, we opted to use the best hyperparameter values as reported in the ViTST paper. The model employs a pre-trained Swin Transformer with a batch size of 48, a learning rate of 0.00002, and a duration of 4 epochs. For handling static data in the P-12 dataset, a pre-trained Roberta-base model is used, consistent with the approach outlined in the original article.

### C.4 Raindrop

Similar to SLAN, the hyperparameters for Raindrop (Zhang et al., 2021), as detailed in Table 3, are determined sequentially in the order listed. Unlike SLAN, Raindrop employs a distinct sampling strategy, as described in the original article. Specifically, sampling involves selecting from a pool consisting of one times the majority class samples and three times the minority class samples, such that every processed batch has the same number of positive and negative class samples.

---

[1] https://github.com/Leezekun/ViTST

### C.5 CoFormer

The hyperparameters for CoFormer (Wei et al., 2023) are listed in Table 3. Due to GPU memory constraints, the batch size is fixed at 16. The remaining parameters, specifically the number of neighbors and the agent encoding dimension, are set to 30 and 32, respectively, aligning with the specifications provided in the original article.

## D  Non-Baseline Models

In this section, we discuss recent models (published in 2023-24) relevant to ISTS data, which were not included as baselines in our study. Notably, GraFITi (Yalavarthi et al., 2024) and Tripletformer (Yalavarthi et al., 2023) are designed for forecasting rather than classification. While it is possible to adapt these forecasting models for classification by using a two-stage process – where the model first imputes the data followed by a classification network predicting outcomes – such an approach could compromise the fairness of comparisons. Therefore, our study limited its scope to models that inherently perform classification tasks. In addition to the above forecasting methods, we could not include the below classification models in our study.

### D.1  ContiFormer (Chen et al., 2024)

The ContiFormer articles detail their outcomes on the MIMIC dataset for event prediction tasks, whereas our study focuses on classification tasks. Although ContiFormer is also suitable for classifying ISTS, as demonstrated in its article across 20 datasets from the UEA Time Series Classification Archive, none of these datasets include MIMIC or P12. For comparison with SLAN, we attempt to apply ContiFormer on the MIMIC and P12 datasets. Due to the complexity of the ContiFormer model, we utilized the original implementation provided by the authors[2]. However, we encountered issues with the code's functionality for classification tasks with MIMIC data. Specifically, the 'forward' function within the 'Physio-Pro/physiopro/model/masktimeseries.py' file assumes the presence of measured values for all sensors at certain timestamps (line 105, 'tmp_mask = torch.bitwise_or(tmp_mask, mask[..., i])'). This assumption does not hold for the MIMIC dataset, leading to implementation failures. Consequently, with its current implementation, ContiFormer is inapplicable for the classification tasks of the MIMIC dataset, and thus we could not include it as a baseline model in our study.

### D.2  TEE4EHR (Karami et al., 2024)

TEE4EHR is designed for classification tasks within EHR datasets, as evidenced by its performance on the P12 dataset. However, we were unable to include this model as a baseline due to the complexity of the data format required by the model. The publication does not offer scripts or detailed guidance on converting raw data into the format suitable for their model. The only reference to data conversion is found on their GitHub page[3], suggesting that one might understand the conversion process by examining one of the processed datasets provided by the authors. Upon reviewing the processed P12 dataset available, the steps required to transform raw data into the final dataset format remained unclear, preventing us from incorporating TEE4EHR as a baseline model in our study.

### D.3  DNA-T (Huang et al., 2024)

The code for DNA-T is not publicly accessible. Consequently, we were unable to include this model as a baseline in our study.

---

[2]https://github.com/microsoft/PhysioPro/tree/main
[3]https://github.com/esl-epfl/TEE4EHR

Table 4: Performance of SLAN when trained on different percentages of training data. The AUPRC and AUROC are reported for Physionet 2012 and MIMIC-III datasets. The metric is reported as the mean ± standard deviation of three runs with different seeds.

| Model | Physionet 2012 | | MIMIC-III | |
|---|---|---|---|---|
| | AUPRC | AUROC | AUPRC | AUROC |
| 25% | 49.51±0.87 | 83.20±0.15 | 46.86±0.78 | 83.84±0.11 |
| 50% | 53.82±0.59 | 85.17±0.31 | 48.85±0.39 | 84.60±0.59 |
| 75% | 54.27±0.42 | 85.88±0.26 | 50.40±0.61 | 85.26±0.53 |
| 100% | 55.20±0.65 | 86.42±0.13 | 51.12±0.57 | 85.63±0.07 |

### D.4   Transformer + TPR (Sun et al., 2024)

The code for this model is available at the GitHub [4]. However, the provided code lacks sufficient detail to facilitate the proper benchmarking of the model. Additionally, the complexity of the model presents significant challenges for implementation. Consequently, we were unable to include this model as a baseline in our study.

## E   Data Scalability

In the practical setting, it is important for any model to have data scalability, meaning the performance of the model on test data should improve as the amount of training data increases. We consider the first 25%, 50%, 75%, and 100% training data for Physionet 2012 and MIMIC-III and train our model on them. The exact values of AUPRC and AUROC are shown in Table 4, and the results are discussed in the section 6 of the main manuscript. The average ± standard deviation of 3 runs is reported here.

---

[4]https://github.com/SCXsunchenxi/TPR

