# OpenReview forum: "No Imputation Needed: A Switch Approach to Irregularly Sampled Time Series"
_TMLR — Rejected by TMLR_

### Review · Reviewer_RFB6 · 2026-03-12

**Summary Of Contributions:**

This paper proposes SLAN (Switch LSTM Aggregate Network), a model for irregularly sampled time series (ISTS) that avoids input-level imputation. The core idea is to assign one LSTM per sensor and use a "switch layer" to activate only those LSTMs whose corresponding sensors are measured at a given timestep. Active LSTMs share information through a global summary state (aggregated cell states) and maintain per-sensor local summary states via Time2Vec-based hidden state decay. The model is evaluated on MIMIC-III and Physionet 2012 for in-hospital mortality prediction.

**Key Strengths:**

- The approach is simple, intuitive, and easy to understand. The per-sensor LSTM + switch mechanism is a clean design.
- Thorough ablation studies covering aggregation functions, concat layer variants, imputation comparisons, data scalability, and robustness to additional missingness (Figure 1).
- Clear presentation with helpful visual illustrations (Figures 2-4) that make the architecture easy to follow.
- Strong results on Physionet 2012 (P-12), outperforming the second-best model by 8.9% AUPRC.

**Key Weaknesses:**

- Only 2 datasets and 1 task type (binary classification for mortality). Generalizability is unclear.
- The "no imputation needed" thesis is undermined by the paper's own results: forward-fill imputed SLAN beats non-imputed SLAN on M-3 AUPRC (51.46 vs 51.12).
- The improvements on M-3 are marginal and within overlapping standard deviations of CoFormer (51.12 +/- 0.57 vs 50.51 +/- 0.90).

**Audience:**

Yes

**Audience Explanation:**

ISTS modeling is an active and important research area, and the clinical application domain is relevant. The simplicity of the approach could appeal to practitioners. However, the limited experimental scope (2 datasets, 1 task) and the incremental nature of the contribution (per-sensor RNNs with selective activation is a relatively straightforward extension of existing ideas like TimeLSTM and Time2Vec) reduce the overall interest level. The paper would be substantially more interesting with broader experimental validation across domains and task types.

**Broader Impact Concerns:**

Since the paper applies to clinical mortality prediction, which is a sensitive domain; and the model is evaluated only on retrospective data, the paper should include a disclaimer that clinical deployment would require prospective validation, regulatory approval, and careful assessment of failure modes, particularly for underrepresented patient populations.

**Claims And Evidence:**

Yes

**Claims Explanation:**

The evidence has several gaps:
1. The central claim that imputation is unnecessary is not consistently supported.

     Table 5 shows forward-fill imputed SLAN achieves higher AUPRC on M-3 (51.46 vs 51.12). The paper downplays this by noting SLAN wins on AUROC and P-12, but this directly contradicts the paper's main thesis on at least one dataset/metric combination.

2. Statistical rigor is insufficient.

   Results are reported over only 3 seeds with no formal significance tests. On M-3, the improvement over CoFormer (51.12 +/- 0.57 vs 50.51 +/- 0.90) has overlapping confidence intervals. The claim of "outperforming" is not statistically sound.

3. The claim that SLAN "eliminates the assumption of any underlying process" is overstated.

    The Time2Vec-based decay (Eq. 5-6) imposes a specific parametric assumption about how hidden state relevance diminishes with time gaps. This is a different assumption than input imputation, but it is still an assumption about the underlying temporal process.

4. The robustness experiment (Figure 1) is convincing but only compares against IPNets. Comparing against the best non-imputation baseline under increased missingness would be more informative.

**Requested Changes:**

1. Evaluate on at least one additional dataset and/or task.

     The P-19 dataset is acknowledged but excluded due to resource constraints. Consider other ISTS benchmarks outside the clinical domain (e.g., activity recognition, climate data) or at minimum a different clinical task (e.g., phenotyping, length-of-stay prediction on the same datasets).

2. Address the forward-fill result honestly.

   The fact that forward-fill-imputed SLAN beats non-imputed SLAN on M-3 AUPRC is a significant finding that contradicts the main thesis. The paper should either: (a) provide a convincing explanation for why this occurs and under what conditions imputation-free modeling is preferred, or (b) soften the claims from "no imputation needed" to a more nuanced position.

3. Add statistical significance tests.

    Use paired t-tests or bootstrap confidence intervals to compare SLAN with the top baselines. Three seeds are a minimum; five would strengthen the results.

---

### Review · Reviewer_XeX1 · 2026-03-23

**Summary Of Contributions:**

**Summary.** SLAN (Switch LSTM Aggregate Network) addresses the bias of imputation in irregularly sampled time series by processing raw observations directly. It uses a dynamic "switch" layer to activate sensor-specific LSTMs only when data is present, maintaining a local summary for each variable and a global summary for cross-sensor context. Validated on MIMIC-III and PhysioNet 2012 for mortality prediction, SLAN outperforms the considered baselines.


**Strengths**

- S1. The paper addresses the critical challenge of modeling irregularly sampled time series (ISTS), a prevalent data structure in numerous industrial and real-world applications.

- S2. The SLAN architecture is logically sound and clearly presented; its novel "switch" mechanism effectively integrates both intra-sensor dynamics and inter-sensor correlations without the need for biased imputation.

- S3. Extensive experiments on two benchmark datasets demonstrate that the proposed method consistently outperforms existing baselines, achieving performance gains in predictive accuracy.


**Weaknesses**

- W1. The experimental validation is restricted to only two clinical datasets. A more diverse set of benchmarks from other domains would strengthen the results. Furthermore, the paper lacks a qualitative ablation study starting from a fully regular dataset. Comparing the model against classical SOTA classification baselines while progressively degrading observation rates would better highlight the specific advantages and performance decay of the SLAN architecture.

- W2. The comparison with existing literature is limited and overlooks several major families of methods specifically designed for irregularly sampled time series. In particular, ODE-based approaches such as [1, 2], Neural/Conditional Process frameworks [3, 4], and more recent implicit neural representation methods for continuous-time modeling [5, 6] are not discussed nor included in the experimental evaluation. These approaches constitute strong and widely recognized baselines for ISTS problems. Their absence makes it difficult to properly position the contribution of the proposed method within the current state of the art.

[1] Rubanova, Y., Chen, R. T. Q., & Duvenaud, D. (2019). Latent ODEs for Irregularly-Sampled Time Series. Advances in Neural Information Processing Systems (NeurIPS).

[2] De Brouwer, E., Simm, J., Arany, A., & Moreau, Y. (2019). GRU-ODE-Bayes: Continuous Modeling of Sporadically-Observed Time Series. NeurIPS.

[3] Garnelo, M., Rosenbaum, D., Maddison, C. J., et al. (2018). Conditional Neural Processes. ICML.

[4] Kim, H., Mnih, A., Schwarz, J., et al. (2019). Attentive Neural Processes. ICLR.

[5] Le Naour E., Serrano L., Migus L.,  et al. (2024). Time Series Continuous Modeling for Imputation and Forecasting with Implicit Neural Representations. TMLR.

[6] Woo, G., Liu, C., Sahoo, D., Kumar, A., & Hoi, S. C. H. (2023). Learning Deep Time-index Models for Time Series Forecasting. ICML.

**Additional Comments:**

None

**Audience:**

Yes

**Audience Explanation:**

Irregularly sampled time series modeling is a highly relevant problem for the time series community, and the proposed architecture appears to be novel and interesting to the best of my knowledge.

**Claims And Evidence:**

No

**Claims Explanation:**

The experimental setup could be strengthened, and the authors should consider discussing additional baselines for irregularly sampled time series modeling.

**Requested Changes:**

Please see weaknesses.

---

### Review · Reviewer_bdCW · 2026-05-04

**Summary Of Contributions:**

This paper proposes SLAN, a Switch LSTM Aggregate Network for irregularly sampled multivariate time series.

Instead of first discretizing the data and imputing missing values, SLAN assigns one TimeLSTM-style recurrent block to each sensor and uses a switch layer to activate only the recurrent blocks corresponding to sensors observed at a given timestamp.

The model maintains per-sensor local hidden summaries, applies hidden state decay using Time2Vec, aggregates active long-term memories into a global summary state, and concatenates final local/global summaries for binary classification.

The paper evaluates SLAN on in-hospital mortality prediction for MIMIC-III and PhysioNet 2012, comparing against imputation-based baselines (GRU-D, IPNets, ViTST) and non-imputation baselines (Transformer, SeFT, Raindrop, CoFormer, IVP-VAE). SLAN achieves the best reported performance on both datasets and both metrics, with especially large AUPRC gains on PhysioNet 2012.


Strength

- The proposed switch-based architecture is simple and effective. It is well matched to the structure of irregular clinical measurements.
- The paper evaluates against a reasonably broad set of imputation and non-imputation baselines.
- The ablations are useful: the paper studies aggregation choices, imputed variants of SLAN, the contribution of local/global summaries, robustness under additional random observation dropping, data-scaling behavior, and sensor-importance patterns.
- The use of AUPRC alongside AUROC is appropriate for the imbalanced mortality-prediction setting.


Weakness

- Some claims are stronger than what the experiments establish. In particular, the claim that imputation is unnecessary or inferior is not fully supported, since the paper only evaluates two clinical classification datasets and an imputed SLAN variant with forward filling slightly outperforms non-imputed SLAN on MIMIC-III AUPRC.
- The comparison protocol raises fairness and reproducibility questions: different sampling strategies are used for some baselines, ViTST is not tuned due to resource constraints, hyperparameter search is sequential rather than joint.
- The complexity/scalability discussion seems optimistic about independence from the number of sensors. Parameter count and memory scale with the number of sensors, and the practical runtime claim depends on GPU parallelism and tensor shapes rather than an architecture-independent complexity result.

**Audience:**

Yes

**Audience Explanation:**

Irregularly sampled time series are widely seen in clinical ML and many other applied settings, and the paper addresses a real modeling problem.

**Broader Impact Concerns:**

The application domain is clinical mortality prediction, so the paper should include a broader-impact discussion even though the contribution is methodological.

**Claims And Evidence:**

No

**Claims Explanation:**

- The title and framing suggest that imputation is generally unnecessary and that avoiding imputation eliminates problematic missingness assumptions. The experiments are narrower: two clinical classification datasets, no forecasting or non-clinical tasks, and no external validation. Moreover, the imputed-SLAN ablation shows that forward-filled SLAN obtains higher MIMIC-III AUPRC than the non-imputed SLAN variant (51.46 vs. 51.12), even though non-imputed SLAN is better on most other reported comparisons. This does not invalidate the method, but it means the evidence supports a more careful claim: SLAN is a promising non-imputation approach that performs strongly on the evaluated mortality-prediction benchmarks.

- I also have concerns about the comparison protocol. Some baselines use different sampling strategies, ViTST is run with reported hyperparameters rather than tuned in the same way because of computational cost, and the hyperparameter search is sequential rather than joint. These choices may be practically reasonable, but they make it harder to attribute the performance gains solely to the proposed modeling idea. The paper also excludes several recent methods for implementation reasons; the appendix explains this, but the resulting comparison is still incomplete relative to the strongest current landscape.

- Finally, there are several clarity and accuracy issues that should be fixed. Most notably, the MIMIC-III instance number appears inconsistent between Table 1 and the text. The time-complexity discussion should more carefully distinguish asymptotic dependence from practical GPU parallelism, since the architecture has one recurrent block per sensor and the parameter count scales linearly with the number of sensors.

- Public code is also important for verifying the reported preprocessing, baseline implementations, and experimental protocol.

**Requested Changes:**

1. Temper the main claims about imputation. The current evidence does not establish that no imputation is generally needed, nor that imputation is broadly inferior. Please revise the title/framing and claims to match the evidence: SLAN is a non-imputation approach that performs strongly on the evaluated benchmarks only.

2. Address the MIMIC-III dataset-count inconsistency. Table 1 reports 22,110 instances, while the dataset description appears to imply 21,110 instances after removing 32 from 21,142. Please correct the count and verify that the reported experiments use the intended split.

3. Clarify baseline fairness. Please explain more explicitly which baselines use different sampling/oversampling strategies, why this is necessary, and whether results change under a common sampling strategy where feasible. Also clarify the implications of not tuning ViTST under the same protocol.

4. Revise the complexity/scalability discussion. The current statement that runtime does not depend on the number of sensors is too strong. Please distinguish theoretical asymptotic scaling, parameter/memory scaling, and practical GPU-parallel runtime. Include empirical runtime/memory measurements as the number of sensors increases, if possible.

---

> ### Comment · Action_Editor_jdx9 · 2026-06-23
> **discussion period and recommendation**
>
> Reviwer,
>
> 1. Please check if the authors have addressed your concerns.
>
> 2.  If not, please comment on their rebuttal.
>
> 3. When appropriate, please submit your recommendation.

---

### Decision · Action_Editor_jdx9 · 2026-06-25

**Recommendation:** Reject

**Audience:**

No

**Audience Explanation:**

Since the authors did not address concerns from the reviewers, the paper is not at a level that could benefit potential readers.

**Claims And Evidence:**

No

**Claims Explanation:**

The authors did not respond to the comments from the reviewers.   Some of the reviewer concerns include claims exceed the provided evidence, only two datasets were used,  more existing methods could have been included for comparison, comparison protocol raises fairness and reproducibility questions, and some improvements are marginal.